# Determinants of incomplete immunization among 12–23 months old children in Ethiopia: A multilevel analysis

**Sofiya Ayalew Kebede[1], Yawkal Tsega (iD)[2]\*, Niguss Cherie[3], Asressie Molla[1], Chad Stecher[4]**

**1** Department of Epidemiology and Biostatistics, School of Public Health, College of Medicine and Health Sciences, Wollo University, Dessie, Ethiopia, **2** Department of Health Systems and Management, School of Public Health, College of Medicine and Health Sciences, Wollo University, Dessie, Ethiopia, **3** Department of Reproductive and Family Health, School of Public Health, College of Medicine and Health Sciences, Wollo University, Dessie, Ethiopia, **4** College of Health Solutions, Arizona State University, Phoenix, Arizona, United States of America

\* yawkaltsega@gmail.com

## Abstract

### Background

Vaccinations saved over 37 million lives between 2000 and 2019. Despite this, Ethiopia's Expanded Program of Immunization has struggled to meet its goals, and little has been studied on the community and individual level determinants of incomplete immunization. Therefore, this study aimed to assess the predictors of incomplete immunization among Ethiopian children aged 12–23 months using Ethiopian Mini Demographic Health Survey 2019 (EMDHS 2019).

### Methods

The study used data from EMDHS 2019 and about 1029 children aged 12–23 months were included in the study. STATA version 17.0 statistical software was used to manage and analyze data. Multilevel binary logistic regression analysis was conducted. An AOR with 95%CI, and $P < 0.05$ were used to determine strength of association and declare significance level, respectively.

### Results

The factors maternal age ranges of 15–24 (AOR: 4.23; 95%CI: 2.17–8.26) and 25–34 (AOR: 2.68; 95%CI: 1.56–4.61), family size ≥5 (AOR: 2.03; 95%CI: 1.24–3.30), ≤3 antenatal care visits (AOR: 2.32; 95%CI: 1.43–3.75), no postnatal care (AOR: 2.20; 95%CI: 1.23–3.95), rural residence (AOR: 2.54; 95%CI: 1.08–6.25), low (AOR=3.55; 95% CI: 1.32–9.55) and moderate (AOR: 3.29; 95% CI: 1.55–7.00) community-level antenatal care services utilization, and low community-level institutional delivery (AOR: 3.93; 95%CI: 1.35–11.50) were the significant determinant factors of incomplete immunization in Ethiopia.

**Data availability statement:** All relevant data are available within the manuscript.

**Funding:** The author(s) received no specific funding for this work.

**Competing interests:** The authors have declared that no competing interests exist.

**Abbreviations:** ANC, antenatal care; BCG, Bacillus Chalmette Guerin; DPT, Diphtheria, Pertussis, Tetanus; EMDHS 2019, Ethiopian Mini Demographic Health Survey 2019; PNC, postnatal care; SDGs, Sustainable Development Goals; SSA, Sub-Saharan Africa; UNICEF, United Nations Children's Fund; WHO, World Health Organization

## Conclusion

Young maternal age, family size, inadequate ANC, rural residence, not utilizing PNC services, and poor wealth status were the individual level determinants of incomplete immunization. Low and moderate level of ANC services utilization, and low community level institutional delivery service utilization were the factors determining incomplete immunization at the community level. Therefore, the health decision makers better to be committed to design strategies to enhance complete immunization coverage and maternal and child health services.

## Introduction

Immunization is one of the most cost-effective health interventions which prevents 2–3 million deaths every year. To safeguard children from vaccine-preventable diseases and improve vaccine access, the World Health Organization (WHO) and the United Nations International Children's Emergency Fund (UNICEF) launched the Expanded Program on Immunization (EPI) in 1977 [1]. By the end of 2021, one in five children worldwide lacked access to essential immunizations, with approximately 25 million children under one year not receiving basic vaccines [2]. The burden of incomplete immunization is not evenly distributed, with low-and middle-income countries are facing the greatest challenges [3,4].

Each year, approximately 6.3 million children under five die globally, with an estimated 1.5 million of these deaths attributable to diseases preventable by immunization [2]. The WHO reported that Sub-Saharan Africa carries higher share of vaccine-preventable deaths due to lower immunization coverage compared to other regions. This region accounts for about half of the 1.5 million vaccine-preventable deaths globally, equivalent to roughly 750,000 deaths annually in Africa [5].

The Ethiopian Demographic Health Survey 2016 (EDHS 2016) stated that only 39% of children aged 12–23 months received all basic vaccinations, and only 22% were fully vaccinated before their first birthday [6]. Likewise, the 2013 immunization report stated that Ethiopia had the second largest number of incompletely vaccinated children in the region, following Nigeria, with overall vaccine coverage at 75% [7]. Moreover, the 2020 UNICEF and WHO report indicated that Ethiopia had approximately 1.1 million children not fully vaccinated, putting them at increased risk for diseases like measles and pneumonia, significant causes of mortality in children under five [8]. Although direct morbidity and mortality rates can vary annually due to outbreaks and vaccination campaigns, over 60% of childhood deaths from vaccine-preventable diseases in Ethiopia are linked to incomplete or missed vaccinations [9].

Several factors such as birth order [10,11], maternal age [10–12], distance from health facilities [11,13], maternal educational status [1,12,13], antenatal care visits [11,14], postnatal care (PNC) visits [14], and delivery site [14,15] contribute to incomplete vaccination coverage.

The WHO member states including Ethiopia aim to achieve the Sustainable Development Goals (SDGs) which aspire to maintain and enhance vaccination efforts to ensure that no child is left behind by 2030 [16]. To reduce incomplete immunization rates, the Ethiopian government has taken several actions including expanding EPI coverage to rural and remote areas [17] and implementing the Health Extension Program (HEP) initiated in 2003, which targets under-vaccinated populations [18]. National Immunization Days (NIDs) were introduced to increase immunization rates, particularly for polio and measles [19].

New vaccines such as the Pneumococcal Conjugate Vaccine (PCV) and the Rotavirus Vaccine have also been introduced [17]. Additionally, mobile health services and outreach

programs have been launched to deliver immunization services to pastoralist and hard-to-reach communities [18]. Ongoing monitoring and surveillance [17] and partnerships with international organizations like Gavi, WHO, and UNICEF help secure funding, technical support, and vaccines (20). Public awareness campaigns are also part of the strategy [18].

However, there is insufficient literature that look into incomplete immunization regarding full-immunization coverage vaccines. Including these vaccines in research aligns with government priorities and national immunization plans [8]. Moreover, understanding the determinants of incomplete childhood immunization is crucial although little has been studied on community level factors of incomplete immunization in Ethiopia [7,20]. Therefore, this study aims to identify the individual and community level factors determining incomplete immunization among children aged 12–23 months using data from the EMDHS 2019.

## Materials and methods

### Study setting, design, period and data source

This study used data from the EMDHS 2019 which is a national representative survey collecting data on various health-related indicators such as fertility, infant and child mortality, reproductive age women's health, and childhood nutritional status in the country. The dataset is available at http://www.dhsprogram.com website. Ethiopia is the second most populous country in Africa and located in the Horn of Africa. Administratively, it has nine regions (Tigray, Afar, Amhara, Oromia, Somali, Benshangul-Gumuz, SNNPR, Gambella and Harar) and two administrative cities (Addis Ababa and DireDawa) in 2019. Ethiopia is a land-locked country located and is bordered by Eritrea, Djibouti, Somalia, Kenya and Sudan and with approximate area of 1,221,900 square kilometers. Moreover, the country's 84% of the total population found in rural areas, while the remaining 16 percent live in urban areas [21,22].

### Population and eligible criteria

All living children aged 12–23 months who were born from reproductive age women in Ethiopia were source population. All alive children aged 12–23 months in Ethiopia who were available during data collection in the selected enumeration areas were the study population.

### Sample size and sampling procedure

A total sample of 1029 children's (weighted), aged 12–23 months were used in Ethiopia. The EMDHS 2019 employs a two-stage stratified probability sampling method. Geographical regions and the urban and rural areas were used to stratify the samples. Using the 2019 Ethiopia Population and Housing Census (EPHC) frame as a basis, 305 EAs (93 in urban and 212 in rural areas) were chosen independently in each sampling stratum with a probability proportionate to their size. In each of the chosen EAs, a household listing operation was conducted. The households on the resulting lists were chosen for the second stage. About 28–30 households were chosen from each cluster using an equal probability systematic selection process in the second stage [9] (Fig 1).

### Variables

**Outcome variable.** The outcome variable for this study was incomplete immunization (categorized as "Yes" and "No"). All basic vaccines were coded as "0" and "1" for those who did not receive and who received the recommended vaccines, respectively. Then all eight recoded variables were add up "0" for fully vaccinated (for who received all eight basic vaccine) and "1" for those who miss at least one of the basic vaccine [23].

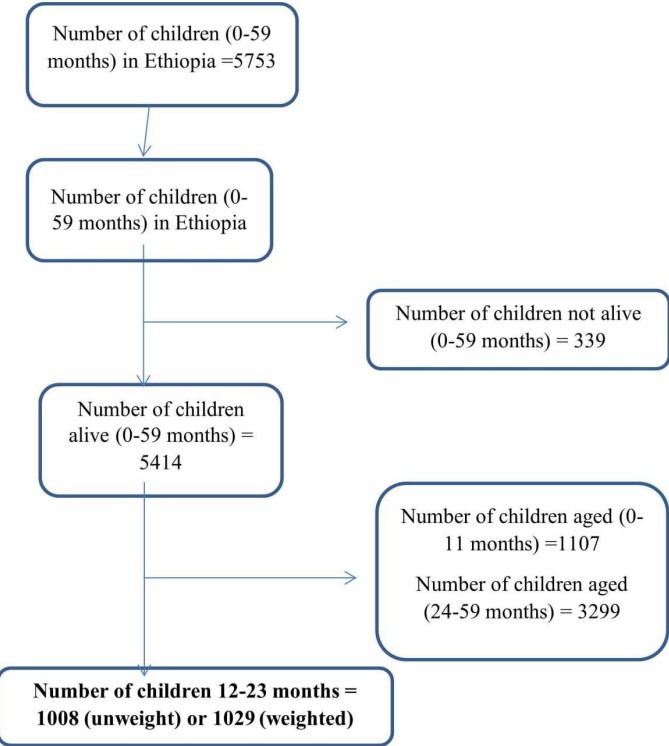

**Fig 1. Pictorial depiction of sampling procedure for the study of determinants of incomplete immunization among 12–23 months old children in Ethiopia, 2019.**

**Explanatory variables.** We extracted additional variables from EMDHS 2019 based on their association in prior studies [13,24]. The variables were categorized into three categories such as socioeconomic and demographic variables (maternal age, education, and autonomy, age and sex of the child, Wealth index, family size, marital status, number of under five children and religion), obstetrical related factors (place of delivery, birth order, number of ANC visits, PNC, parity and breast feeding), and informational related factors (media exposure and health information from health provider).

**Community level factors.** Residence, season of birth, Ethiopian administrative regions, the community level maternal education, community level wealth index, community level media exposure, community level ANC utilization and community level institutional delivery were variables directly extracted from EMDHS 2019 and generated by aggregating the individual observation in a cluster. The aggregates were computed using the proportion of the given variables subcategory. Since the aggregated value for all generated variables not normally distributed it was categorized in to groups as low, moderate and high based on previous related studies [25–27].

## Operational definitions

*Media exposure.* A frequency of listening to the radio and watching television were used to measure media exposure in this study. So women exposure to either television or radio at least once a week was considered exposed, otherwise not exposed [28].

**Community-level media usage.** Community level media usage was an aggregate respondent level of exposure for different type of media categorized as "<25%=low", "25%-50%=moderate" and ">50%= high media utilized communities" [25].

**Community-level women education.** Community level women's education was defined as the percentage of reproductive age mothers in the community with primary or higher education (categorized as "≤ 25%=low", "25%-75%=moderate" and "> 75%= high education communities") [25].

**Community level wealth status.** Community wealth index was defined as the proportion of the households in the community in the upper 40% wealth quintile (richer and richest) (categorized as < 25%=low, 25%-50%=moderate and > 50%= high wealth status communities) [29].

**Community level ANC utilization.** The proportion of mothers with in specific clusters who visited ANC for some number of times. It was categorized using national level quartiles in to low utilized community (when ≤25% of women are utilizing ANC), middle (when 25%-75% of women are utilizing ANC), and high (when > 75% of women are utilizing ANC) [25].

**Community level institutional delivery.** Proportion of mothers who had hospital delivery in the community and categorized as low (<25%), middle (25%-50%) and high (>50%) [30].

## Data processing and analysis

*Multilevel binary logistic regression.* STATA version 17.0 was used to manage and analyze data. The sample was weighted (v005/1000000) to ensure that the survey results are representative of the population and used for analysis and descriptive statistics. A multilevel binary logistic regression model was used to determine the individual and community-level factors affecting incomplete immunization. Variables with a $P < 0.2$ in bi-variable logistic regression analysis was included for multivariable analysis [31]. Then a multivariable multilevel binary logistic regression model was used to estimate the determinants of incomplete immunization. Moreover, multi-collinearity among the explanatory variables was checked using the variance inflation factor (VIF), if VIF is greater than 10 considered as there is a multi-collinearity [32]. In multivariable analysis, a $P<0.05$ was considered statistically significant and interpreted with Adjusted Odds Ratio (AOR) and 95% confidence interval.

Then a total of 4 models were fitted; the first is the null model without any explanatory variables were developed to evaluate the null hypothesis that there is no cluster-level difference in incomplete immunization that specified only random interest. The likelihood ratio test (LR), intra class correlation coefficient (ICC) (to see the proportion of total variance attributable to the community level, the higher the ICC (ICC >10%) the more relevant the community characteristics for understanding individual variation in incomplete immunization) [27], and it was calculated using the following formula:

$$ICC = \frac{V_A}{V_A + \frac{\pi^2}{3}}$$ , Where VA is community-level variance, and $\pi^2/3 = 3.29$ is individual-level variance.

On the other hand, we checked proportional change of variance (PCV) and median odds ratio (MOR) (to quantify the variation between communities by comparing two respondents from two randomly chosen communities) was computed to measure the variation (random effect) by the following formula to determine the fitness of the model:

$$PCV = \frac{Vo - Vi}{Vo}$$ , Where Vo is a variance in the null model, and Vi is a variance in the consecutive model.

$$MOR = e^{0.95 V_i}$$ , Were Vi indicates the cluster variance.

The second is for individual-level explanatory variables, the third is for community-level explanatory variables, and finally both individual and community-level explanatory variables

simultaneously. The model comparison was done using deviance information criteria (DIC), Bayesian Information Criterion (BIC), log-likelihood test, and Akaike information criteria (AIC). Since the model is nested. A model with the highest log likelihood test and lowest DIC, BIC, and AIC was selected for reporting and interpreting results.

## Ethical considerations

No ethical approval was required for this study as we utilized the publicly available demographic and health survey (DHS) data, which is anonymized before release. We obtained an authorization letter to download the DHS dataset from the Central Statistical Agency (CSA) after submitting a request through https://dhsprogram.com/. The dataset and all methodologies employed in this study adhered to the principles outlined in the Declaration of Helsinki and were conducted per DHS research guidelines.

## Results

### Socioeconomic and demographic characteristics

A total of 1029 children aged 12–23 months were included in this study. The mean ± standard deviation (SD) of the age of children was 17.02 ± 3.47 months. About 519 (50.46%) of children's mother were in the age group of 25–34 years from those 295 (56.79%) of mothers did not completely immunize their children. Regarding mother's educational status, 464 (45.16%) and 418 (40.61%) of the participants had no and primary education, respectively among those 310 (66.76%) and 208 (49.90%) mothers did not completely immunize their child respectively. Among the participants 431 (41.89%) had poor wealth index among those 282 (65.58%) had incompletely immunized. Coming to family size, households that had greater than or equal to five were 644 (62.64%) from those 384 (59.68%) of them were not completely immunized their children (Table 1).

### Obstetric related factors

Regarding to antenatal care visit, about 439 (42.67%) children whose mothers attended ≥4 ANC visit from those 172 (39.23%) of them were incompletely immunize. Coming to postnatal care visit, 893 (86.86%) of children did not get postnatal care among those 526 (58.94%) were incompletely immunize. More over 552 (53.67%) of mothers who had 3 and above number of parities among those 332 (60.08%) of their children were incompletely immunized (Table 2).

### Informational related factors

About 629 (61.14%) of mothers were not exposed to media of which 382 (60.68%) mothers did not completely immunize their child. Coming to gaining health information from health care provider 512 (49.81%) children whose mothers did not get health information from health care provider among those 355 (69.29%) were incompletely immunized.

### Community level factors

As shown in Table 3 below 715 (69.53%) of children live in rural area from those 457 (63.85%) were incompletely immunized. Regarding to community level maternal education 212 (20.54%) of children who live in low community level maternal education among those 139 (65.55%) were incompletely immunized. Coming to community level antenatal care utilization 246 (23.92%) of children whose mothers have low antenatal care utilization from those 194.83 (79.23%) were incompletely immunized (Table 3).

**Table 1. Socioeconomic and demographic characteristics of respondents and study children aged 12–23 months in Ethiopia, EMDHS 2019.**

| Variables | Frequency | Percent | Incomplete immunization | |
|---|---|---|---|---|
| | | | Yes | No |
| | | | Freq. (%) | Freq. (%) |
| Maternal age | | | | |
| 15–24 | 309.35 | 30.09 | 186.33 (60.23) | 123.02 (39.77) |
| 25–34 | 518.78 | 50.46 | 294.59 (56.79) | 224.19 (43.21) |
| ≥35 | 200.02 | 19.45 | 93.94 (46.97) | 106.08 (53.03) |
| Maternal education | | | | |
| No education | 464.32 | 45.16 | 309.99 (66.76) | 154.32 (33.24) |
| Primary | 417.53 | 40.61 | 208.35 (49.90) | 209.18 (50.10) |
| Secondary | 84.59 | 8.23 | 34.95 (41.32) | 49.64 (58.68) |
| Higher | 61.71 | 6.00 | 21.57 (34.94) | 40.15 (65.06) |
| Maternal autonomy | | | | |
| Yes | 141.45 | 13.76 | 76.71 (54.23) | 64.75 (45.77) |
| No | 886.70 | 86.24 | 498.15 (56.18) | 388.54 (43.82) |
| Age of the child in month | | | | |
| 12–14 | 304.54 | 29.62 | 156.48 (51.38) | 148.06 (48.62) |
| 15–17 | 256.46 | 24.94 | 146.96 (57.30) | 109.50 (42.70) |
| 18–20 | 242.01 | 23.54 | 137.24 (56.71) | 104.77 (43.29) |
| 21–23 | 225.14 | 21.90 | 134.18 (59.60) | 90.96 (40.40) |
| Wealth index | | | | |
| Poor | 430.69 | 41.89 | 282.43 (65.58) | 148.26 (34.42) |
| Middle | 178.62 | 17.37 | 115.44 (64.63) | 63.18 (35.37) |
| Rich | 418.84 | 40.74 | 176.99 (42.26) | 241.85 (57.74) |
| Sex of the child | | | | |
| Male | 495.24 | 48.17 | 270.13 (54.55) | 225.11 (45.45) |
| Female | 532.91 | 51.83 | 304.73 (57.18) | 228.18 (42.82) |
| Family size | | | | |
| ≤4 | 384.13 | 37.36 | 190.51 (49.60) | 193.62 (50.40) |
| ≥5 | 644.02 | 62.64 | 384.35 (59.68) | 259.67 (40.32) |
| Marital status | | | | |
| Married | 983.12 | 95.62 | 547.50 (55.69) | 435.62 (44.31) |
| Unmarried | 45.03 | 4.38 | 27.36 (60.77) | 17.67 (39.23) |
| Religion | | | | |
| Orthodox | 383.25 | 37.28 | 161.07 (42.03) | 222.18 (57.97) |
| Muslim | 350.47 | 34.09 | 209.01 (59.64) | 141.46 (40.36) |
| Other* | 294.43 | 28.64 | 204.78 (69.55) | 89.65 (30.45) |
| No under 5 children in the house hold | | | | |
| ≤1 | 430.70 | 41.89 | 210.95 (48.98) | 219.75 (51.02) |
| 2 | 500.09 | 48.64 | 283.21 (56.63) | 216.87 (43.37) |
| ≥3 | 97.36 | 9.47 | 80.70 (82.88) | 16.67 (17.12) |

*Protestant, catholic and traditional.

## Prevalence of incomplete immunization coverage

The prevalence of incomplete immunization among children aged 12–23 month was 567 (56.25%) (95% CI: 53.12–59.34). There were a regional variation in the prevalence of

**Table 2. Obstetric factors for the study of incomplete immunization and its associated factors among children aged 12–23 months in Ethiopia, EMDHS 2019.**

| Variables | Frequency | Percentage | Incomplete immunization | |
|---|---|---|---|---|
| | | | Yes | No |
| | | | Freq. (%) | Freq. (%) |
| Number of ANC visit | | | | |
| No visit | 283.23 | 27.55 | 222.14 (78.43) | 61.10 (21.57) |
| Three or less | 306.22 | 29.78 | 180.62 (58.98) | 125.60 (41.02) |
| Four and above | 438.70 | 42.67 | 172.10 (39.23) | 266.59 (60.77) |
| PNC visit | | | | |
| Yes | 135.13 | 13.14 | 48.54 (35.92) | 86.59 (64.08) |
| No | 893.02 | 86.86 | 526.32 (58.94) | 366.70 (41.06) |
| Parity | | | | |
| 1 | 230.28 | 22.40 | 121.79 (52.89) | 108.49 (47.11) |
| 2 | 246.02 | 23.93 | 121.51 (49.39) | 124.51 (50.61) |
| ≥3 | 551.85 | 53.67 | 331.56 (60.08) | 220.29 (39.92) |
| Breast feeding | | | | |
| Yes | 802.72 | 78.07 | 407.15 (50.72) | 395.57 (49.28) |
| No | 225.43 | 21.93 | 167.71 (74.40) | 57.72 (25.60) |
| Place of delivery | | | | |
| Home | 464.26 | 45.15 | 338.89 (73.00) | 125.37 (27.00) |
| Health facility | 563.89 | 54.85 | 235.97 (41.85) | 327.92 (58.15) |
| Birth order | | | | |
| Fist | 241.90 | 23.53 | 131.81 (54.49) | 110.09 (45.51) |
| 2–3 | 382.14 | 37.17 | 193.33 (50.59) | 188.81 (49.41) |
| 4–5 | 215.89 | 21.00 | 130.56 (60.48) | 85.33 (39.52) |
| >5 | 188.22 | 18.31 | 119.16 (63.31) | 69.06 (36.69) |

incomplete immunization, higher in Afar, Oromia and SNNPR, whereas the lowest proportion of incomplete immunization was found in Addis Ababa, Tigray and Benshangul (Fig 2).

## Factors affecting child incomplete immunization

Variables with $P < 0.2$ in bivariable analysis were selected for multivariable analysis. Those variables with $P<0.05$ in multivariable analysis reported as a significant determinant of incomplete immunization usage. Bivariable analysis was done to assess any relation between each independent variables and incomplete immunization coverage. During the assessment of Multicollinearity, we did variance inflation factor (VIF) test for all explanatory variables <10 ranging from 1.06 to 3.53.

## The random effects analysis result

In multilevel multivariate analysis, model 1 (individual-level factors), model 2 (community-level factors), and model 3 (both individual and community-level factors) were considered. The ICC value was computed for each model, the ICC value in the null model was 0.51, indicating that due to variability between clusters, and there was a 51.4% variation in incomplete immunization. The between-cluster variability declined over successive models, for model 1 it was 42.6% and for model 3, the ICC value was 35.3% indicating variability in clusters. The proportional change in variance indicates that the addition of predictors to the empty models explained an increased proportion of variance in incomplete immunization. Similar to the ICC value the combined model showed PCV value, i.e., 48.36% of the variance

**Table 3. Community-level factors for the study of incomplete immunization and its associated factors among children age 12–23 months old in Ethiopia, EMDHS 2019.**

| Variables | Frequency | Percentage | Incomplete immunization | |
|---|---|---|---|---|
| | | | Yes | No |
| | | | Frequency (%) | Frequency (%) |
| Regions | | | | |
| Tigray | 77.41 | 7.53 | 20.89 (26.99) | 56.52 (73.01) |
| Afar | 15.00 | 1.46 | 12.06 (80.35) | 2.95 (19.65) |
| Amhara | 217.85 | 21.19 | 80.91 (37.14) | 136.94 (62.86) |
| Oromia | 405.22 | 39.41 | 287.04 (70.83) | 118.19 (29.17) |
| Somali | 55.88 | 5.43 | 45.55 (81.52) | 10.33 (18.48) |
| Benshangul | 10.66 | 1.04 | 3.61 (33.88) | 7.05 (66.12) |
| SNNPR | 199.23 | 19.38 | 112.57 (56.50) | 86.66 (43.50) |
| Gambella | 4.15 | 0.40 | 2.49 (60.18) | 1.65 (39.82) |
| Harari | 2.55 | 0.25 | 1.41 (55.30) | 1.14 (44.70) |
| Addis Ababa | 34.18 | 3.32 | 5.71 (16.72) | 28.47 (83.28) |
| DireDawa | 6.02 | 0.59 | 2.62 (43.56) | 3.39 (56.44) |
| Residence | | | | |
| Urban | 313.32 | 30.47 | 118.42 (37.80) | 194.90 (62.20) |
| Rural | 714.83 | 69.53 | 456.44 (63.85) | 258.39 (36.15) |
| Season of birth | | | | |
| Dry | 266.48 | 25.92 | 169.46 (63.59) | 97.02 (36.41) |
| Winter | 285.43 | 27.76 | 143.42 (50.25) | 142.01 (49.75) |
| Spring | 261.26 | 25.41 | 144.50 (55.31) | 116.76 (44.69) |
| Rainy | 214.98 | 20.91 | 117.48 (54.65) | 97.50 (45.35) |
| Community-level maternal education | | | | |
| Low | 211.17 | 20.54 | 138.43 (65.55) | 72.75 (34.45) |
| Moderate | 583.27 | 56.73 | 338.99 (58.12) | 244.27 (41.88) |
| High | 233.71 | 22.73 | 97.44 (41.69) | 136.27 (58.31) |
| Community-level wealth index | | | | |
| Low | 382.33 | 37.19 | 249.40 (65.23) | 132.93 (34.77) |
| Moderate | 127.91 | 12.44 | 93.88 (73.40) | 34.03 (26.60) |
| High | 517.91 | 50.37 | 231.58 (44.71) | 286.33 (55.29) |
| Community-level media exposure | | | | |
| Low | 331.88 | 32.28 | 200.40 (60.38) | 131.48 (39.62) |
| Moderate | 144.15 | 14.02 | 91.06 (63.17) | 53.09 (36.83) |
| High | 552.12 | 53.70 | 283.40 (51.33) | 268.72 (48.67) |
| Community-level ANC utilization | | | | |
| Low | 245.90 | 23.92 | 194.83 (79.23) | 51.07 (20.77) |
| Moderate | 411.23 | 40.00 | 275.49 (66.99) | 135.74 (33.01) |
| High | 371.02 | 36.09 | 104.54 (28.18) | 266.48 (71.82) |
| Community-level institutional delivery | | | | |
| Low | 252.00 | 24.51 | 202.16 (80.22) | 49.84 (19.78) |
| Moderate | 387.45 | 37.68 | 231.12 (59.65) | 156.33 (40.35) |
| High | 388.70 | 37.81 | 141.58 (36.42) | 247.12 (63.58) |

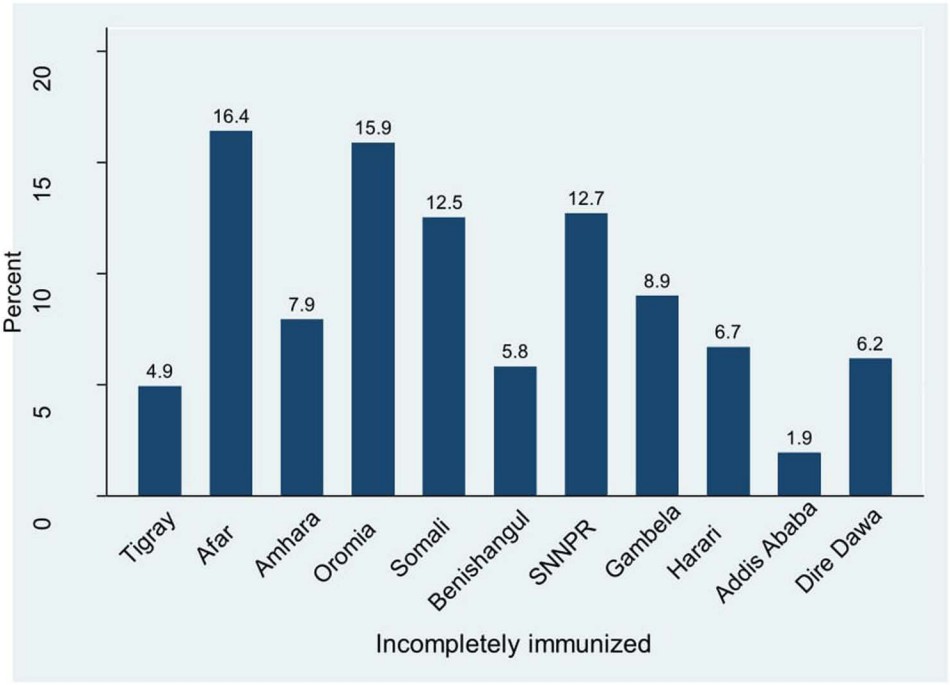

**Fig 2. Magnitude of incomplete immunization across Ethiopian administrative regions, EMDHS 2019.**

in incomplete immunization could be explained by the combined factors at the individual and community level.

The MOR for incomplete immunization was 27.27 in the empty model, which indicated the presence of variation between communities (clustering) in incomplete immunization since MOR was 27 times higher than the reference (MOR=1). The unexplained community variation in incomplete immunization decreases to 5.51 (MOR=5.51) when all factors were added to the empty model. This implies that even though individual and community-level factors were considered, the effect of clustering is still statistically significant in the final model (see Table 2). As shown in Table 2, the model fit statistics, the value of higher log-likelihood and the lowest AIC value in model 3 is determined as the best fitted model. The final model identified 7 significant variables (maternal age, family size, number of ANC visits, PNC, residence, community ANC utilization, and community institutional delivery) of incomplete immunization (Table 4).

## Fixed effect (measures of association) results

Multilevel multivariable logistic regression analysis (Table 4 below) showed that children whose mothers ages 15–24 and 25–34 were 4.23 (AOR= 4.23, 95% CI: 2.17–8.26) and 2.68 (AOR=2.68, 95%CI: 1.56–4.61) times higher odds of incomplete immunization usage compared to those who born from mothers age greater than or equal to 35 years respectively. The odds of incomplete immunization were 2.03 (AOR= 2.03, 95%CI=1.24–3.30) times higher among children who came from large families (≥ 5) than children who came from small families (≤4). Children born from mothers who had no antenatal care and ≤ three antenatal care visits had 2 (AOR=2, 95% CI: 1.03–3.89) and 2.32 (AOR=2.32, 95%CI: 1.43–3.75) times more likely to be incompletely immunized than mothers who had ≥ 4 antenatal care visit

Table 4. Random effect (measures of variations) results on the factors associated with incomplete immunization among children aged 12–23 months in Ethiopia, EMDHS 2019.

| Random effect | Null model | Model 1 | Model 2 | Model 3 |
|---|---|---|---|---|
| Variance | 3.48 (2.29–5.29) | 2.44(1.53–3.90) | 1.67 (1.01–2.77) | 1.80 (1.07–3.03) |
| ICC | 0.5140 | 0.4260 | 0.3369 | 0.3533 |
| MOR | 27.27 | 10.17 | 4.90 | 5.51 |
| PCV | Ref | 0.2983 | 0.5197 | 0.4836 |
| LR test | 258.10 ($P$ = 0.000) | 130.00 ($P$ = 0.000) | 88.45 ($P$ = 0.000) | 76.00 ($P$ = 0.000) |
| Wald Chi-square | | 116.03 | 86.34 | 132.53 |
| Model fitness | | | | |
| Log-likelihood | -576.40 | -503.69 | -521.62 | -478.80 |
| AIC | 1156.81 | 1055.38 | 1079.23 | 1037.61 |
| BIC | 1166.64 | 1173.36 | 1167.72 | 1234.23 |
| DIC | 1152.8 | 1007.38 | 1043.24 | 957.6 |
| N | 1,008 | 1,008 | 1008 | 1008 |
| Number of groups | 286 | 286 | 286 | 286 |

LR test: likelihood ratio test, ICC: Intra-class Correlation Coefficient, MOR: Median Odds Ratio, PCV: Proportional Change of Variance, AIC: Akaike's Information Criterion, BIC: Bayesians Information Criterion, DIC: Deviance Information Criteria, N = sample size.

respectively. The odds of incomplete immunization were 2.20 times higher among children born from women who had PNC follow-up than their encounter parts (AOR = 2.20, 95% CI: 1.23–3.95). Children who live in rural residence were 2.59 times more likely to be incompletely immunized than children who live in urban area (AOR=2.59, 95% CI: 1.08–6.25). Moreover, children residing in communities with low and moderate proportion of ANC utilization had 3.55 times (AOR = 3.55, 95% CI: 1.32–9.55) and 3.29 times higher odds (AOR = 3.29, 95% CI: 1.55–7.00) of experiencing incomplete immunization compared with children residing in communities with high proportion of ANC utilization respectively. Children who came from low community level institutional delivery had 3.93 (AOR = 3.93, 95% CI: 1.35–11.50) times high likely incompletely immunized than children came from high community level institutional delivery (Table 5).

## Discussion

This study aimed to determine the determinants of incomplete immunization among children aged 12–23 months in Ethiopia using multilevel analysis of EMDHS 2019. Younger maternal age, no ANC visit, no PNC visit of the mother, large family size, rural residence, community-level antenatal care visit, and community-level institutional delivery were the significant factors for incomplete immunization. Among the associated factors, younger maternal age was positively linked with incomplete immunization, consistent with findings from various other studies [21,33,34]. This suggests that maternal maturity is crucial in achieving higher childhood vaccination coverage compared to early marriages. Therefore, childbearing at a younger age and the age at first birth require careful consideration. One possible explanation for this finding is that young mothers often lack experience in child care. Over time, older mothers have acquired essential skills and knowledge in child care, making them more informed about child health practices and more likely to utilize services that improve their child's health [34].

The study also found that the likelihood of incomplete immunization was higher among children whose mothers did not attend antenatal care. This finding aligns with previous studies in Pakistan [33] and Nigeria [35]. On the other hand, children residing in communities

**Table 5. Individual and community level determinants of incomplete immunization among children age 12–23 months in Ethiopia, EMDHS 2019.**

| Variables | Model 0 | Model 1 | Model 2 | Model 3 |
|---|---|---|---|---|
| | | AOR (95% CI) | AOR (95% CI) | AOR (95% CI) |
| Maternal age | | | | |
| 15–24 | | 4.75 (2.45–9.20)*** | | 4.23 (2.17–8.26)*** |
| 25–34 | | 2.84 (1.66–4.85)*** | | 2.68 (1.56–4.61)*** |
| 35 and above | | 1 | | 1 |
| Age of the child | | | | |
| 12–14 | | 1 | | 1 |
| 15–17 | | 1.34 (0.80–2.24) | | 1.18 (0.61–2.27) |
| 18–20 | | 0.90 (0.53–1.52) | | 0.98 (0.46–2.09) |
| 21–23 | | 0.98 (0.55–1.72) | | 1.15 (0.58–2.30) |
| Maternal education | | | | |
| No education | | 1.84 (0.66–5.12) | | 1.68 (0.57–4.92) |
| Primary | | 0.57 (0.21–1.50) | | 0.54 (0.20–1.46) |
| Secondary | | 1.76 (0.64–4.83) | | 1.58 (0.56–4.45) |
| Higher | | 1 | | 1 |
| Wealth index | | | | |
| Poor | | 2.14 (1.15–4.01)* | | 1.72 (0.86–3.47) |
| Middle | | 2.26 (1.19–4.29) * | | 1.96 (0.98–3.90) |
| Rich | | 1 | | 1 |
| Family size | | | | |
| ≤4 | | 1 | | 1 |
| ≥5 | | 1.93 (1.19–3.13) ** | | 2.03 (1.24–3.30) * |
| No of under-five children | | | | |
| ≤1 | | 1 | | 1 |
| 2 | | 1.06 (0.68–1.64) | | 1.05 (0.67–1.63) |
| ≥3 | | 2.00 (0.86–4.66) | | 2.04 (0.86–4.86) |
| Religion | | | | |
| Orthodox | | 1 | | 1 |
| Muslim | | 2.38 (1.19–4.79) * | | 1.66 (0.78–3.52) |
| Others * | | 2.90 (1.41–5.97)** | | 1.98 (0.96–4.09) |
| ANC visit | | | | |
| No visit | | 2.46 (1.30–4.65) ** | | 2.00 (1.03–3.89) * |
| Three or less | | 2.52 (1.57–4.07)*** | | 2.32 (1.43–3.75) ** |
| Four and above | | 1 | | 1 |
| PNC visit | | | | |
| Yes | | 1 | | 1 |
| No | | 2.05 (1.15–3.67)* | | 2.20 (1.23–3.95) ** |
| Breast feeding | | | | |
| Yes | | 1 | | 1 |
| No | | 1.41 (0.84–2.37) | | 1.35 (0.79–2.29) |
| Place of delivery | | | | |
| Home | | 1.89 (1.14–3.12) * | | 1.25 (0.72–2.17) |
| Health facility | | 1 | | 1 |
| Media exposure | | | | |

*(Continued)*

**Table 5.** (Continued)

| Variables | Model 0 | Model 1 | Model 2 | Model 3 |
|---|---|---|---|---|
| | | AOR (95% CI) | AOR (95% CI) | AOR (95% CI) |
| Yes | | 1 | | 1 |
| No | | 0.84 (0.50–1.41) | | 0.97 (0.56–1.69) |
| Health information from health providers | | | | |
| Yes | | 1 | | 1 |
| No | | 0.82 (0.49–1.35) | | 0.76 (0.45–1.28) |
| Regions | | | | |
| Agrarian | | | 1 | 1 |
| Pastoralist | | | 4.28 (1.49–12.32) ** | 2.56 (0.78–8.43) |
| City dweller | | | 0.77 (0.23–2.60) | 0.72 (0.20–2.66) |
| Residence | | | | |
| Urban | | | 1 | 1 |
| Rural | | | 3.16 (1.39–7.21) ** | 2.59 (1.08–6.25) * |
| Season of birth | | | | |
| Dry | | | 1.53 (0.95–2.47) | 1.20 (0.62–2.32) |
| Winter | | | 1 | 1 |
| Spring | | | 1.23 (0.76–1.99) | 1.04 (0.56–1.96) |
| Rainy | | | 0.92 (0.55–1.52) | 0.75 (0.35–1.62) |
| Community-level maternal education | | | | |
| Low | | | 1.23 (0.45–3.35) | 0.66 (0.20–2.14) |
| Moderate | | | 1.14 (0.54–2.44) | 0.83 (0.36–1.92) |
| High | | | 1 | 1 |
| Community-level wealth index | | | | |
| Low | | | 1.38 (0.61–3.10) | 1 |
| Moderate | | | 3.44 (1.07–11.07) * | 1.05 (0.41–2.66) |
| High | | | 1 | 3.04 (0.89–10.38) |
| Community-level media exposure | | | | |
| Low | | | 0.25 (0.11–0.58) ** | 0.25 (0.10–2.64) |
| Moderate | | | 0.24 (0.09–0.65) ** | 0.19 (0.07–1.57) |
| High | | | 1 | 1 |
| Community-level ANC utilization | | | | |
| Low | | | 4.82 (1.97–11.80) ** | 3.55 (1.32–9.55) * |
| Moderate | | | 3.58 (1.76–7.26) *** | 3.29 (1.55–7.00) ** |
| High | | | 1 | 1 |
| Community institutional delivery | | | | |
| Low | | | 5.32 (2.15–13.15) *** | 3.93 (1.35–11.50)* |
| Moderate | | | 2.33 (1.15–4.74) * | 1.97 (0.87–4.46) |
| High | | | 1 | 1 |

AOR= Adjusted Odds Ratio, *=$P$<0.05, **=$P$<0.01, ***=$P$ <0.001, 1= reference category.

with a low proportion of ANC utilization were also a risk for incomplete immunization. However, this finding is not significant in another previous study [20], possibly due to the influence of other factors, differences in categorization, and variations in sample size. The reason may be that attending antenatal clinics provides mothers with valuable information about

vaccination and other child health practices. During these visits, individualized counseling and health education sessions help mothers understand the importance of vaccination, build confidence in the health services provided, encourage institutional delivery, and improve child nutrition uptake [35].

When assessing community-level factors, residence was associated with incomplete immunization. Children living in rural areas were more likely to have incomplete immunizations, as confirmed by previous studies in Nigeria [35] and Ethiopia [21]. This disparity may be due to greater awareness of the importance of immunization, more available resources, and a higher number of health facilities in urban areas, which are more accessible. In contrast, rural communities face higher travel costs to reach vaccination sites [21,36].

The likelihood of incomplete immunization was higher for children born to mothers from communities with low institutional delivery rates compared to those from communities with high institutional delivery rates. This may be because mothers who give birth in health facilities benefit from the initial vaccinations given at birth and receive education on the importance of completing their children's vaccinations [35]. However, this variable was not significant in another study [20], possibly due to the influence of other factors, differences in categorization, and variations in sample size.

Another study also indicated a significant variation in predictors of incomplete immunization across different geographical regions [33,37]. The research findings highlight a positive correlation between larger family sizes (≥ 5 members) and incomplete immunization which is supported by previous studies conducted in Australia [38] and the Yirga Chefie district of Southern Ethiopia [39]. Possible reasons for this consistency include resource constraints, limited access to care for older children, parental attention being divided among multiple children, familial stress, lack of transportation, and time constraints [40].

Poor household wealth status had a significant association with incomplete immunization. This finding was supported by other previous studies in Punjab, Pakistan[41], Brazil [42], Bangladeshi [12], Nigeria [35,36], South Africa [43], Togo [13], Yirga Chefie district of south Ethiopia [39] and Bale zone of Ethiopia [44]. This might be due to that impoverished families often to visit the healthcare, including free immunization services. Financial limitations prevent access to healthcare facilities, secondary expenses, such as buying vaccination records and obtaining medication for vaccine-related healthcare. Moreover, household poverty presents obstacles to ensuring children receive vaccinations, resulting in poor nutrition, weakened immunity, and vulnerability to diseases due to neglecting vaccination [13,35].

In addition mothers who did not receive PNC exhibited a significant association with incomplete immunization, aligning with findings from various studies in Low-income and Middle Income Countries (LMICs). These studies illustrate that enhanced health communication regarding immunization during Maternal and Child Health (MCH) visits is closely linked to childhood immunization rates [36,45]. This finding were supported by previous research conducted in Nigeria [36] and Ethiopia [24]. This could be due to that attending clinics for PNC enables mothers to receive comprehensive information on vaccination, other child health practices, mothers undergo personalized counseling and health education sessions that raise awareness about vaccination, boost confidence in available health services, and enhance childhood vaccination rates [35,39].

## Limitation of the study

As we used secondary data, important explanatory variables like transportation services, distance from the health care facility, and availability of vaccine, paternal education and occupation are not incorporated. Additionally, this study used data from cross-sectional study design, which is difficult to determine temporal association between outcome variable and the

explanatory variables. This survey interviewed women who have a child born 3 year before the survey, recall bias is inevitable.

## Conclusions

The rate of immunization coverage remains below the WHO target. Variables such as maternal age, number of ANC visit, family size, PNC visit, place of residence, community level ANC utilization and community level institutional delivery were factors determining incomplete immunization. Prioritized intervention should be given for younger mothers, rural residence, for poor household, large family members and give emphasis for maternal and child health.

## Recommendations

To address the determinants of incomplete immunization among children aged 12–23 months in Ethiopia, it is crucial to implement policies that promote maternal education and delay early marriages, enhance access to antenatal and postnatal care services, and improve healthcare infrastructure in rural areas. Community-based health programs should be strengthened to increase awareness and utilization of immunization services, while socioeconomic barriers must be addressed through financial support and social protection programs. Practical measures include launching educational campaigns, engaging community leaders, providing incentives for health visits, and collaborating with NGOs and international organizations to leverage resources and expertise. By focusing on these targeted interventions, Ethiopia can improve immunization coverage and ensure better health outcomes for children.

## Acknowledgments

The authors are honored to appreciate the Demographic Health Survey Program for providing EMDHS 2019 datasets with authorization letter.

## Author contributions

**Conceptualization:** Sofiya Ayalew Kebede, Yawkal Tsega, Niguss Cherie, Asressie Molla, Chad Stecher.

**Data curation:** Sofiya Ayalew Kebede, Yawkal Tsega.

**Formal analysis:** Sofiya Ayalew Kebede, Yawkal Tsega, Asressie Molla, Chad Stecher.

**Funding acquisition:** Niguss Cherie.

**Investigation:** Sofiya Ayalew Kebede, Yawkal Tsega, Asressie Molla, Chad Stecher.

**Methodology:** Sofiya Ayalew Kebede, Yawkal Tsega, Niguss Cherie, Asressie Molla, Chad Stecher.

**Project administration:** Sofiya Ayalew Kebede, Yawkal Tsega, Niguss Cherie, Asressie Molla.

**Resources:** Sofiya Ayalew Kebede, Yawkal Tsega, Niguss Cherie, Asressie Molla.

**Software:** Sofiya Ayalew Kebede, Yawkal Tsega, Asressie Molla, Chad Stecher.

**Supervision:** Yawkal Tsega, Niguss Cherie, Asressie Molla, Chad Stecher.

**Validation:** Sofiya Ayalew Kebede, Yawkal Tsega, Niguss Cherie, Asressie Molla, Chad Stecher.

**Visualization:** Sofiya Ayalew Kebede, Yawkal Tsega, Niguss Cherie, Asressie Molla, Chad Stecher.

**Writing – original draft:** Sofiya Ayalew Kebede, Yawkal Tsega.

**Writing – review & editing:** Sofiya Ayalew Kebede, Yawkal Tsega, Niguss Cherie, Asressie Molla, Chad Stecher.

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
