## [Decision Letter · Decision Letter 0]

13 Jan 2025

PONE-D-24-57676Determinants of incomplete immunization among 12-23 months old children Ethiopia: A Multilevel analysisPLOS ONE

Dear Dr. Tsega,

Thank you for submitting your manuscript to PLOS ONE. After careful consideration, we feel that it has merit but does not fully meet PLOS ONE’s publication criteria as it currently stands. Therefore, we invite you to submit a revised version of the manuscript that addresses the points raised during the review process.

We look forward to receiving your revised manuscript.

Kind regards,

Kahsu Gebrekidan, Ph.D.

Academic Editor

PLOS ONE

Journal Requirements:

3. We note you have included a table to which you do not refer in the text of your manuscript. Please ensure that you refer to Table 1 and 5 in your text; if accepted, production will need this reference to link the reader to the Table.

Reviewers' comments:

Reviewer's Responses to Questions

**Comments to the Author**

1. Is the manuscript technically sound, and do the data support the conclusions?

Reviewer #1: Yes

Reviewer #2: Yes

Reviewer #3: Yes

2. Has the statistical analysis been performed appropriately and rigorously? 

Reviewer #1: Yes

Reviewer #2: Yes

Reviewer #3: Yes

3. Have the authors made all data underlying the findings in their manuscript fully available?

Reviewer #1: Yes

Reviewer #2: Yes

Reviewer #3: No

4. Is the manuscript presented in an intelligible fashion and written in standard English?

Reviewer #1: Yes

Reviewer #2: No

Reviewer #3: No

5. Review Comments to the Author

Reviewer #1: In the introduction section, it is explained that Ethiopia has a high percentage of incomplete vaccines, perhaps short-term or long-term targets for immunization coverage in Ethiopia can be explained as a reference.

Several "(Error! Reference source not found.)" texts were found that need to be corrected so that citations can refer to the appropriate references.

In the results section, related to the limitations mentioned, it is possible to explain recommendations for further research so as not to encounter similar limitations that can be in line with the recommendations from the research results in the conclusion section.

In the conclusion section, it is explained that prioritized intervention should be given for younger mothers, rural residence, for poor household, large family members and give emphasis for maternal and child health, perhaps specific recommendation for the government or Ethiopian health decision makers can be mentioned in line with the research results.

Reviewer #2: Manuscript id: PONE-D-24-57676

The study titled "Determinants of incomplete immunization among 12-23 months old children Ethiopia: A Multilevel analysis"

General Comment:

Immunization is a significant public health success; however, countries classified as low-income and lower-middle-income continue to face challenges in achieving full immunization coverage. This paper identifies determinants at both the individual and community levels, which are of great importance.

Authors could elaborate on how this study adds value to the existing knowledge.

I am recommending publishing this manuscript with some major changes.

Specific Comments:

Introduction

• Line 63 and 64: “Each year….. by immunization”. This could be moved up in the first paragraph.

• Follow the sequence – Globally, Regional and Country; Incomplete immunization, Deaths…..

• Line 96: word “long-term coverage” could be replaced by “fully-immunization”.

• Line 99: Please add references

Materials and Methods

• Line 115, 116,117: There is repetition. Please check.

• I am concerned about the study design; this is secondary analysis. Please indicate in the design. The description showed as if it was a survey (primary data collection).

• The quality of the figures could be improved.

Results: Overall, result tables and writeup require re-formatting and revision. (e.g Sentence case, P value either p-value, p value, P value- same formatting throughout the document, etc)

• (Error! Reference source not found.) Mentioned at multiple places in the manuscript.

• Table 1 heading needs revision

• Table 2 heading needs revision

• Line 269: “usage” – “coverage”

• Table 5 not readable.

• Obstetrical related factor could be included in “individual factors”

Discussion:

• Reference 43 – provide Doi

• Line 384: What is MGWR software?

• The last paragraph of the discussion section is well written

Conclusion:

• Please re-write the conclusion.

• Further, there are no recommendations pulled out from this analysis.

• Authors could add some recommendations at individual and community level to overcome the immunization challenges in Ethiopia and in countries with similar problems.

Reviewer #3: Thanks! for PLOS ONE editor office for giving me a chance to review this research article titled “Determinants of incomplete immunization among 12-23 months old children Ethiopia:

A Multilevel analysis”

General comment to authors:

This work is valid from a scientific standpoint, technically sound, and original to the literature.

On the other hand, the work has received some minor editing, and other corrections.

• Despite Ethiopia's lack of recent EDHS, it would be beneficial to get the most recent data on the factors that contributing to incomplete immunization currently, as we are now in 2025,those factors in 2019 were might be reduced or eliminated.

Abstract part

Line 35-36: 3 or more antenatal care (ANC) visits, you mean it is significant factor to have incomplete immunization? It isn’t clear.

Conclusion part

Line 40: Since the aforementioned notion runs counter to your finding on inadequate ANC, may we conclude that three or more ANC follow-ups were insufficient/inadequate? And better if you add recommendation too.

Main body of the Manuscript

Materials and methods part

Line 107-109: Despite the fact that the prior administrative system is crucial to this investigation, it is best to describe the current one as well.

Line 184-185: “if VIF is greater than 10 considered as there is multi-collinearity” What is the numerical result of this study? Write it if possible.

Line 228,238, 253, 292, 299,318: “(Error! Reference source not found.)” is not clear. If you have missed references here insert it please.

Result and Discussion Part

It is well-structured, discussed, and scientifically sound.

Lastly, since this poll was conducted at the national level, it would be preferable if you include recommendation as country level.

6. PLOS authors have the option to publish the peer review history of their article (what does this mean? ). If published, this will include your full peer review and any attached files.

**Do you want your identity to be public for this peer review?** For information about this choice, including consent withdrawal, please see our Privacy Policy .

Reviewer #1: No

Reviewer #2: No

Reviewer #3: **Yes: ** Alemu Bogale (MD,MPH)

---

## [Author Response · Author response to Decision Letter 1]

26 Feb 2025

Date: 26 February 2025

To: PLOS ONE

Subject: Submission of Revised Manuscript

Dear Editor and Reviewers,

We appreciate your feedbacks to our manuscript # PONE-D-24-57676, titled " Determinants of incomplete immunization among 12-23 months old children in Ethiopia: A Multilevel analysis". Your valuable comments, and expert suggestions have significantly enhanced the quality of our manuscript.

In light of the constructive feedbacks, we have meticulously revised and updated the manuscript. Furthermore, an English language expert has reviewed the manuscript to correct any grammatical inaccuracies.

We are eager to publish the manuscript in your reputable journal, PLOS ONE, to reach a relevant audience and influence policy changes aimed at eradicating vaccine preventable disease in Ethiopia and other similar settings

Once again, we would like to reiterate our profound gratitude to the editor and reviewers for their time and constructive feedback. We have provided responses to all comments point by point below.

Best regards,

Yawkal Tsega

Corresponding author

On behalf of the authors

Email:yawkaltsega@gmail.com

Mobile: +251933559351

Response to Reviewer 1 comments

Comment 1: In the introduction section, it is explained that Ethiopia has a high percentage of incomplete vaccines, perhaps short-term or long-term targets for immunization coverage in Ethiopia can be explained as a reference.

Authors’ response: Dear Reviewer 1, we are grateful for your expert comments, suggestions and recommendations. We tried to revise the introduction section to be more comprehensive ad clearer in the revised manuscript through taking your comments into consideration.

Comment 2: Several "(Error! Reference source not found.)" texts were found that need to be corrected so that citations can refer to the appropriate references.

Authors’ response: Thank you so much for bringing this point to our attention. It was the citation error occurred during word to PDF conversion and we revised it in the revised manuscript.

Comment 3: In the results section, related to the limitations mentioned, it is possible to explain recommendations for further research so as not to encounter similar limitations that can be in line with the recommendations from the research results in the conclusion section.

Authors’ response: We appreciate your insightful suggestions. We revised the recommendation part as per your suggestion.

Comment 4: In the conclusion section, it is explained that prioritized intervention should be given for younger mothers, rural residence, for poor household, large family members and give emphasis for maternal and child health, perhaps specific recommendation for the government or Ethiopian health decision makers can be mentioned in line with the research results.

Authors’ response: Thank you once again for your careful review of our work. We revised the conclusion section both in the abstract and the main body of the manuscript.

Response to Reviewer 2 comments

General comments

Comment 1: Immunization is a significant public health success; however, countries classified as low-income and lower-middle-income continue to face challenges in achieving full immunization coverage. This paper identifies determinants at both the individual and community levels, which are of great importance. Authors could elaborate on how this study adds value to the existing knowledge. I am recommending publishing this manuscript with some major changes.

Authors’ response: Dear Reviewer 2, Thank you so much for your positive evaluation and kind and encouraging words. We have meticulously revised the manuscript based on yours and other reviewers’ comments, suggestions, and recommendations.

Introduction

Comment 2: Line 63 and 64: “Each year…..by immunization”. This could be moved up in the first paragraph.

Authors’ response: We appreciate your insight. We have moved this statement above to keep the logical flow of the introduction.

Comment 3: Follow the sequence – Globally, Regional and Country; Incomplete immunization, Deaths…..

Authors’ response: Thank you so much for bringing this point to our attention. We have restructured the introduction to keep the logical flow of the based on your suggestions. .

Comment 4: Line 96: word “long-term coverage” could be replaced by “fully-immunization”.

Authors’ response: We revised based on your comment.

Comment 5: Line 99: Please add references

Authors’ response: We added references in the revised manuscript.

Materials and Methods

Comment 6: Line 115, 116,117: There is repetition. Please check.

Authors’ response: We checked it up again this part and we tried to address the repetitions in the revised manuscript

Comment 7: I am concerned about the study design; this is secondary analysis. Please indicate in the design. The description showed as if it was a survey (primary data collection).

Authors’ response: Thank you for your detailed comments. We value your comments and suggestions and we revised this section in the revised manuscript.

Comment 8: The quality of the figures could be improved.

Authors’ response: Thank you so much for your insights. We tried to improve the qualities of the figures in the revised manuscript.

Results:

Comment 9: Overall, result tables and write up require re-formatting and revision. (e.g Sentence case, P value either p-value, p value, P value- same formatting throughout the document, etc)

Authors’ response: Thank you so much for your insightful comments. We revised the typos other issues in all tables in the revised manuscript.

Comment 10: (Error! Reference source not found.) Mentioned at multiple places in the manuscript.

Authors’ response: Thank you so much for bringing this point to our attention. It was the citation error occurred during word to PDF conversion and we revised it in the revised manuscript.

Comment 11: Table 1 heading needs revision

Authors’ response: We revised the table 1 title based on your comment.

Comment 12: Table 2 heading needs revision

Authors’ response: We revised the table 2 title based on your comment.

Comment 13: Line 269: “usage” – “coverage”

Authors’ response: We replaced usage by coverage based on your suggestion.

Comment 14: Table 5 not readable.

Authors’ response: We revised the table 5 in the revised manuscript.

Comment 15: Obstetrical related factor could be included in “individual factors”

Authors’ response: We appreciate your insight. We presented these factors under separated subtopic and table to let the readers to have emphasis.

Discussion

Comment 16: Reference 43 – provide Doi

Authors’ response: Thank you so much. We add the doi for this reference.

• Comment 17: Line 384: What is MGWR software?

Authors’ response: We apologise for the typological error. We revised it in the revised manuscript.

Comment 18: The last paragraph of the discussion section is well written

Authors’ response: Thank you so much for your encouraging words.

Conclusion

Comment 19: Please re-write the conclusion.

Authors’ response: We appreciate your insight and we updated the conclusion in the revised manuscript.

Comment 20: Further, there are no recommendations pulled out from this analysis.

Authors’ response: We valued your suggestions and we add the recommendations in the revised manuscript.

Comment 21: Authors could add some recommendations at individual and community level to overcome the immunization challenges in Ethiopia and in countries with similar problems.

Authors’ response: We valued your suggestions and we add the recommendations in the revised manuscript.

Response to Reviewer 3 comments

Comment 1: Thanks! for PLOS ONE editor office for giving me a chance to review this research article titled “Determinants of incomplete immunization among 12-23 months old children Ethiopia: A Multilevel analysis”

Authors’ response: Dear Reviewer 3, we appreciate you for your time and exert comments and suggestions you provided on our work.

General comment to authors:

Comment 2: This work is valid from a scientific standpoint, technically sound, and original to the literature.

Authors’ response: Thank you so much for your positive evaluation. Your comments and suggestions were highly instrumental for the enhancement of the quality of the manuscript.

Comment 3: On the other hand, the work has received some minor editing, and other corrections.

Authors’ response: We tried to address the comments and took into account your recommendations in the revised manuscript.

Comment 4: Despite Ethiopia's lack of recent EDHS, it would be beneficial to get the most recent data on the factors that contributing to incomplete immunization currently, as we are now in 2025,those factors in 2019 were might be reduced or eliminated.

Authors’ response: Thank you for your thoughtful insight. We agree that the factors are crucial factors although we could not incorporate them in the current study due to the secondary data nature.

Abstract part

Comment 5: Line 35-36: 3 or more antenatal care (ANC) visits, you mean it is significant factor to have incomplete immunization? It isn’t clear.

Authors’ response: Thank you for your comment. To clarify, the statement indicates that having three or fewer antenatal care (ANC) visits is a significant risk factor for incomplete immunization when compared to the reference category of mothers who have four or more ANC visits. This means that a lower number of ANC visits increases the risk of incomplete immunization, highlighting the importance of ensuring that mothers attend at least four ANC visits to improve immunization coverage for their children.

Comment 6: Conclusion part: Line 40: Since the aforementioned notion runs counter to your finding on inadequate ANC, may we conclude that three or more ANC follow-ups were insufficient/inadequate? And better if you add recommendation too.

Authors’ response: Thank you so much for your thought provoking insights. We revised the conclusion part of the abstract in the revised manuscript.

Materials and methods part

Comment 7: Line 107-109: Despite the fact that the prior administrative system is crucial to this investigation, it is best to describe the current one as well.

Authors’ response: Dear Editor,

Comment 8: Line 184-185: “if VIF is greater than 10 considered as there is multi-collinearity” What is the numerical result of this study? Write it if possible.

Authors’ response: Thank you so much for your comment. Yes, you are right writing the VIF result was great, however, by the time of analysis we conduct VIF analysis and exclude variables with VIF>10 from the analysis.

Comment 9: Line 228,238, 253, 292, 299,318: “(Error! Reference source not found.)” is not clear. If you have missed references here insert it please.

Authors’ response: Thank you so much for bringing this point to our attention. It was the citation error occurred during word to PDF conversion and we revised it in the revised manuscript.

Result and Discussion Part

Comment 10: It is well-structured, discussed, and scientifically sound.

Authors’ response: We are so grateful for your kind and encouraging g words.

Comment 11: Lastly, since this poll was conducted at the national level, it would be preferable if you include recommendation as country level.

Authors’ response: Once again, we would like to appreciate you for your time, positive and expert evaluations on our manuscript. We added the recommendation section in the revised manuscript.

---

## [Decision Letter · Decision Letter 1]

10 Mar 2025

PONE-D-24-57676R1Determinants of incomplete immunization among 12-23 months old children in Ethiopia: A Multilevel analysisPLOS ONE

Dear Dr. Tsega,

Thank you for submitting your manuscript to PLOS ONE. After careful consideration, we feel that it has merit but does not fully meet PLOS ONE’s publication criteria as it currently stands. Therefore, we invite you to submit a revised version of the manuscript that addresses the points raised during the review process.

We look forward to receiving your revised manuscript.

Kind regards,

Kahsu Gebrekidan, Ph.D.

Academic Editor

PLOS ONE

Journal Requirements:

Reviewers' comments:

Reviewer's Responses to Questions

**Comments to the Author**

1. If the authors have adequately addressed your comments raised in a previous round of review and you feel that this manuscript is now acceptable for publication, you may indicate that here to bypass the “Comments to the Author” section, enter your conflict of interest statement in the “Confidential to Editor” section, and submit your "Accept" recommendation.

Reviewer #2: All comments have been addressed

Reviewer #3: All comments have been addressed

2. Is the manuscript technically sound, and do the data support the conclusions?

Reviewer #2: Yes

Reviewer #3: Yes

3. Has the statistical analysis been performed appropriately and rigorously? 

Reviewer #2: Yes

Reviewer #3: Yes

4. Have the authors made all data underlying the findings in their manuscript fully available?

Reviewer #2: Yes

Reviewer #3: Yes

5. Is the manuscript presented in an intelligible fashion and written in standard English?

Reviewer #2: Yes

Reviewer #3: Yes

6. Review Comments to the Author

Reviewer #2: Immunization is a significant public health success; however, countries classified as low-income and lower-middle-income continue to face challenges in achieving full immunization coverage. This paper identifies determinants at both the individual and community levels, which are of great importance.

Authors addressed all comments appropriately.

I am recommending publishing this manuscript.

Reviewer #3: I want to thank the authors for taking the time to answer all of my main concerns. I appreciate that, but I could have one more.;

Abstract part

Comment 5: Line 35-36: 3 or more antenatal care (ANC) visits, you mean it is significant

factor to have incomplete immunization? It isn’t clear.

Authors’ response: "Thank you for your comment. To clarify, the statement indicates that

having three or fewer antenatal care (ANC) visits is a significant risk factor for incomplete

immunization when compared to the reference category of mothers who have four or more

ANC visits. This means that a lower number of ANC visits increases the risk of incomplete

immunization, highlighting the importance of ensuring that mothers attend at least four ANC

visits to improve immunization coverage for their children."

I got you in this case it cannot contradict with existing knowledge, or disparities biological plausibility. But your abstract result on both line 35 and on first page of PDF, still says,"≥3 antenatal care

visits (AOR: 2.32; 95%CI: 1.43-3.75)" which contradicts with your response, please use the sign "≤" rather than ≥ according to your response.

7. PLOS authors have the option to publish the peer review history of their article (what does this mean? ). If published, this will include your full peer review and any attached files.

**Do you want your identity to be public for this peer review?** For information about this choice, including consent withdrawal, please see our Privacy Policy .

Reviewer #2: No

Reviewer #3: **Yes: ** Alemu Bogale MD,MPH

---

## [Author Response · Author response to Decision Letter 2]

11 Mar 2025

Date: 11 March 2025

To: PLOS ONE

Subject: Submission of Revised Manuscript

Dear Editor and Reviewers,

We are grateful for the commitment, feedbacks and reviews on our manuscript # PONE-D-24-57676R1, titled " Determinants of incomplete immunization among 12-23 months old children in Ethiopia: A Multilevel analysis". Your detailed and careful comments are really substantial to enhance the quality of the manuscript.

In accordance with the reviewers comments and the journal requirements, we tried to address all the raised issues diligently. In light of the constructive feedbacks, we have meticulously revised and updated the manuscript. Furthermore, an English language expert has reviewed the manuscript to correct any grammatical inaccuracies and we did not cite any retracted articles in this manuscript.

We have keen interest to publish the manuscript in your reputable journal, PLOS ONE, to reach a relevant audience and influence policy changes aimed at eradicating vaccine preventable disease in Ethiopia and other similar settings.

Once again, we would like to appreciate the editor and reviewers for their time and constructive feedback. We have provided responses to all comments point by point below.

Best regards,

Yawkal Tsega

Corresponding author

On behalf of the authors

Email:yawkaltsega@gmail.com

Mobile: +251933559351

Response to Reviewer 2 comments

Comment 1: Authors addressed all comments appropriately. I am recommending publishing this manuscript.

Authors’ response: Dear Reviewer 2, we are highly thankful for your positive evaluation of our paper. Your detailed revision has been highly instrumental to improve the quality of the paper.

Response to Reviewer 3 comments

Comment 5: Line 35-36: 3 or more antenatal care (ANC) visits, you mean it is significant factor to have incomplete immunization? It isn’t clear. I got you in this case it cannot contradict with existing knowledge, or disparities biological plausibility. But your abstract result on both line 35 and on first page of PDF, still says,"≥3 antenatal care visits (AOR: 2.32; 95%CI: 1.43-3.75)" which contradicts with your response, please use the sign "≤" rather than ≥ according to your response.

Authors’ response: Dear Reviewer 3, we appreciate your suggestions and for bringing this point to our attention. We revised it as your recommendation in the revised the manuscript.

---

## [Editor Report · Decision Letter 2]

14 Mar 2025

Determinants of incomplete immunization among 12-23 months old children in Ethiopia: A Multilevel analysis

PONE-D-24-57676R2

Dear Mr. Yawkal,

We’re pleased to inform you that your manuscript has been judged scientifically suitable for publication and will be formally accepted for publication once it meets all outstanding technical requirements.

Kind regards,

Kahsu Gebrekidan, Ph.D.

Academic Editor

PLOS ONE
---

## [Editor Report · Acceptance letter]

PONE-D-24-57676R2

PLOS ONE

Dear Dr. Tsega,

I'm pleased to inform you that your manuscript has been deemed suitable for publication in PLOS ONE. Congratulations! Your manuscript is now being handed over to our production team.

Kind regards,

on behalf of

Dr. Kahsu Gebrekidan

Academic Editor

PLOS ONE